# MetaMath: Bootstrap Your Own Mathematical Questions for Large Language Models

**Longhui Yu[1,\*]    Weisen Jiang[2,3,\*]    Han Shi[4,†]    Jincheng Yu[3,4]    Zhengying Liu[4]**
**Yu Zhang[2]    James T. Kwok[3]    Zhenguo Li[4]    Adrian Weller[1,5]    Weiyang Liu[1,6,†]**

[1]University of Cambridge    [2]Southern University of Science and Technology
[3]Hong Kong University of Science and Technology    [4]Huawei Noah's Ark Lab
[5]The Alan Turing Institute    [6]Max Planck Institute for Intelligent Systems - Tübingen

yulonghui@stu.pku.edu.cn, wjiangar@cse.ust.hk, shi.han@huawei.com, wl396@cam.ac.uk

Project page: **meta-math.github.io**

## Abstract

Large language models (LLMs) have pushed the limits of natural language understanding and exhibited excellent problem-solving ability. Despite the great success, most existing open-source LLMs (*e.g.*, LLaMA-2) are still far away from satisfactory for solving mathematical problems due to the complex reasoning procedures. To bridge this gap, we propose *MetaMath*, a finetuned language model that specializes in mathematical reasoning. Specifically, we start by bootstrapping mathematical questions by rewriting the question from multiple perspectives, which results in a new dataset called MetaMathQA. Then we finetune the LLaMA-2 models on MetaMathQA. Experimental results on two popular benchmarks (*i.e.*, GSM8K and MATH) for mathematical reasoning demonstrate that MetaMath outperforms a suite of open-source LLMs by a significant margin. Our MetaMath-7B model achieves $66.5\%$ on GSM8K and $19.8\%$ on MATH, exceeding the state-of-the-art models of the same size by $11.5\%$ and $8.7\%$. Particularly, MetaMath-70B achieves an accuracy of $82.3\%$ on GSM8K, slightly better than GPT-3.5-Turbo. We release the MetaMathQA dataset, the MetaMath models with different model sizes and the training code for public use.

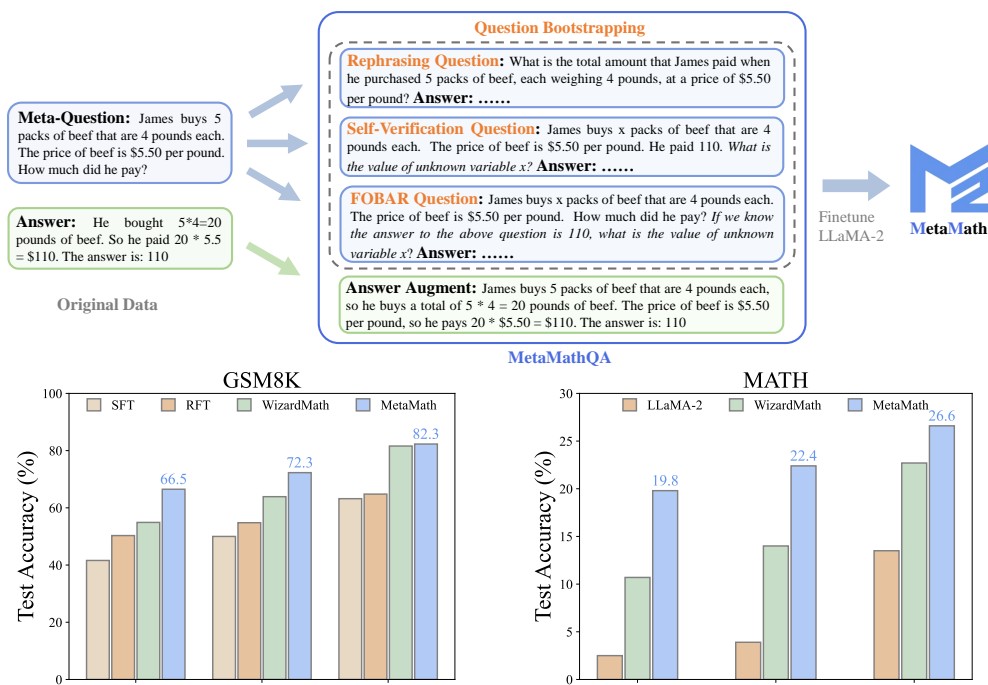

Figure 1: Overview of the MetaMathQA dataset and the mathematical problem-solving LLM – MetaMath. We note that our MetaMath-70B is finetuned by QLoRA [15] due to the computing resource limitation.

---

*Equal contribution    †Corresponding author

# 1 INTRODUCTION

Recent years have witnessed the rapid development of large language models (LLMs) which emerge as the favored approach for various applications and demonstrate multi-dimensional abilities, including instruction following [7, 38, 55, 67], coding assistance [8, 36, 44, 51], and mathematical problem-solving [14, 28, 43, 79]. Among various tasks, solving mathematical problems is more challenging as they often require highly complex and symbolic multi-step reasoning capabilities. Although some close-sourced models, *e.g.*, GPT-3.5-Turbo [52], GPT-4 [54] and PaLM-2 [70], have demonstrated promising performance on some mathematical problem-solving benchmarks, it is still a mystery how these models are trained and what data these models use. Therefore, how to equip open-source LLMs (*e.g.*, LLaMA [69, 70]) with good mathematical problem-solving skills remains an open challenge.

To tackle this challenge, two popular lines of research to improve the mathematical problem-solving abilities of LLMs are: *prompt-based methods* and *finetuning-based methods*. Prompt-based methods [20, 74, 75, 77, 78, 84] aim to activate the potential capacities of LLMs by choosing suitable prompting inputs without modifying the model parameters. Finetuning-based methods update the open-source LLMs (*e.g.*, LLaMA) under the guidance of some other powerful closed-source LLMs (*e.g.*, GPT-3.5 [52], GPT-4 [54]). While prompt-based methods are model-dependent and sensitive to many factors, finetuning-based methods, despite being simple and model-agnostic, heavily rely on effective training data on downstream mathematical questions. Our work aims to improve finetuning-based methods with a novel method to bootstrap available mathematical questions in the training set. Specifically, we propose to bootstrap the questions in both forward and backward reasoning directions. For the forward direction, we have the original and LLM-rephrased questions. For the backward direction, we have the self-verification question [76] and FOBAR question [32]. To construct backward reasoning questions, we mask a token in a question using an identifier "x" and ask the model to predict the masked token if the answer is provided. Different from [32, 76] that apply backward reasoning for inference verification, we use it as a form of question for language model fine-tuning. For answers, we adopt an answer augmentation method based on rejection sampling [79], where diverse reasoning paths are generated and only those with correct answers are used. After combining both forward and backward mathematical questions with augmented answers, we construct a new dataset for fine-tuning, called *MetaMathQA*. By fine-tuning LLaMA-2 on MetaMathQA, we obtain our *MetaMath* model. Our approach is guided by the insight that a mathematical question represents merely a single view of the underlying meta-knowledge. Therefore, question bootstrapping can be viewed as a form of multi-view augmentation in order to enable the transfer of the meta-knowledge. Leveraging the MetaMathQA dataset, MetaMath demonstrates exceptional performance in mathematical reasoning, positioning it among the top performers on widely recognized evaluation benchmarks.

Another motivation behind question bootstrapping is to enlarge the question diversity [18] such that the question distribution can be rich enough to cover more unseen scenarios. We quantify the question diversity of the original questions and our MetaMathQA dataset in Figure 2. The diversity gain [6] indicates how diverse the question is compared to the existing dataset, and a larger diversity gain means the new question is more different from the existing dataset. With question bootstrapping, our Meta-MathQA dataset is much more diverse than the original dataset. We also observe that the test accuracy without bootstrapped questions rapidly reaches a state of saturation. In contrast, the test accuracy, when using bootstrapped questions, continues to exhibit a steady increase.

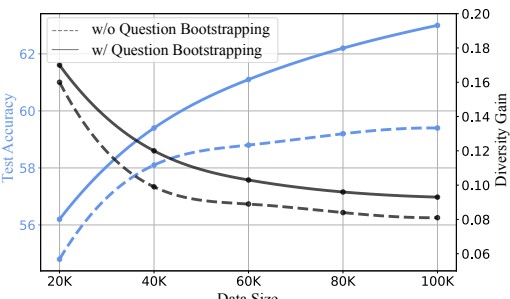

Figure 2: GSM8K accuracy of LLaMA-2-7B finetuned on different sizes of answer augmentation data. A larger diversity gain indicates the question is more diverse compared to the existing questions. Detailed experimental setup is given in Section 4.1.

Question bootstrapping also has an intrinsic connection to dataset distillation [73, 82] and machine teaching [40, 41, 58, 85], where the shared target is to construct a training dataset that best facilitates generalization. Unlike both methods that focus on optimizing the training empirical risk, question bootstrapping uses the reasoning diversity of questions as a heuristic proxy and maximizes this diversity by constructing forward, backward and rephrased questions. MetaMath aims to transfer the underlying meta-knowledge to enable strong generalization [34]. Our contributions are listed below:

- We propose a novel question bootstrapping method to augment the training dataset, resulting in MetaMathQA. Question bootstrapping rewrites questions with both forward and backward reasoning paths and also leverages LLMs to rephrase the question text.

- Based on the MetaMathQA dataset, MetaMath is finetuned from state-of-the-art open-source LLMs (*e.g.*, LLaMA-2), showing excellent elementary mathematical problem-solving capability.

- We identify an important factor when creating the MetaMathQA dataset – question diversity. The diversity is particularly important in reasoning directions, and backward reasoning questions are very helpful for LLMs to understand mathematical knowledge without memorization.

- We conduct experiments on two standard mathematical reasoning benchmarks: GSM8K [13] and MATH [23]. MetaMath outperforms existing open-source LLMs by a large margin. MetaMath-7B has achieved 66.5% on GSM8K (+11.5% compared to the previous best open-source LLM) on GSM8K and 19.8% on MATH (+8.7% compared to the previous best open-source LLM).

- Our work studies data augmentation for improving the mathematical problem-solving ability of LLMs. Despite being simple, our method significantly outperforms many intricate methods. Our results highlight the importance of data augmentation and also shed light on other reasoning tasks.

## 2 RELATED WORK

**Large Language Models (LLMs)** [7, 16, 42, 59, 60, 65, 69] have achieved great success in various natural language processing tasks, *e.g.*, topic classification [31, 33, 47], sentiment classification [7, 47], translation [7], by few-shot prompting (or in-context learning) [7, 10, 47]. Recently, Wang et al. [74], Wei et al. [75] show that LLMs with more than 100B parameters (*e.g.*, GPT-3 [7] with 175B, PaLM with 540B [12]) can solve complex tasks by generating multiple reasoning steps towards the answer when given a few reasoning examples as demonstration. While both GPT-3.5 [52] and GPT-4 [54] have shown promising reasoning ability for complex mathematical tasks like MATH [23], the performance of open-source models (*e.g.*, LLaMA-1 [69], LLaMA-2 [70]) is far from satisfactory.

**Learning Mathematical Reasoning** for complex math tasks like GSM8K [13] and MATH [23] is one of the most challenging problem in open-source LLMs. Wei et al. [75] enhances the reasoning ability of LLMs by augmenting the output with a sequence of intermediate steps toward the answer. A few methods [20, 74, 84] are proposed to improve the quality of reasoning paths. For example, Complexity-based CoT [20] selects examples with more steps as in-context demonstrations and shows that prompting with more reasoning steps leads to better performance. Self-Consistency [74] samples multiple reasoning paths and selects the final answer by majority voting. Another category of work is finetuning-based methods, which finetunes open-source models (*e.g.*, LLaMA) with the knowledge from some advanced closed-source LLMs [52, 54]. Magister et al. [45] investigates the transfer of reasoning capabilities via knowledge distillation. Yuan et al. [79] proposes to apply rejection sampling finetuning (RFT) to improve mathematical reasoning performance. WizardMath [43] proposes a reinforced evol-instruct method to enhance reasoning abilities by supervised fine-tuning and PPO training [62]. MAmmoTH [80] combines CoT and Program-of-Thought [9] rationales for teaching LLMs to use external tools (e.g., Python interpreter) for solving mathematical problems. Wang et al. [72] propose a constraint alignment loss to finetune LLMs for calibration.

**Knowledge Distillation** [21, 24] transfers knowledge from a larger teacher model to a smaller student model, achieving promising performance in many applications [22, 48, 56, 63], Recently, [19, 25–27, 37, 45, 64] propose to transfer reasoning abilities from LLMs (*e.g.*, GPT-3.5 [52], PaLM [12]) to small language models (*e.g.*, T5 [60], GPT-2 [59]). For example, Finetune-CoT [25] samples multiple reasoning paths from LLMs and finetune the student model with correct ones, while Self-Improve [27] chooses the one with the highest confidence. Li et al. [37] further feeds the question and ground-truth label to LLMs for prompting its reasoning path. Shridhar et al. [64] proposes to generate sub-questions and solution pairs for training. Small models finetuned by knowledge distillation can achieve similar performance to LLMs [25, 45] on both common sense reasoning (*e.g.*, CommonSenseQA [66]) and symbol reasoning (*e.g.*, Coin Flip [75]). However, for solving challenging mathematical problems (*e.g.*, GSM8K [13]), there is still a large performance gap [19, 25, 45].

## 3 METHOD

The overview of our method is illustrated in Figure 1. Given a meta-question (a sample in the original mathematical training set), we can generate a series of variants. Specifically, we perform three types of question bootstrapping. Combined with answer augmentation, we present MetaMathQA, a diverse

and high-quality mathematical dataset based on GSM8K and MATH. We then present MetaMath, a family of LLMs finetuned on MetaMathQA focusing on elementary mathematical problem-solving.

## 3.1 ANSWER AUGMENTATION (ANSAUG)

Generating more reasoning paths is a simple but effective way to augment the training set. For a question $q_i$, we use few-shot chain-of-thought prompting with temperature sampling to generate $K_{\text{AnsAug}}$ more reasoning paths $\{(r_i^{(j)}, a_i^{(j)}) : j = 1, \ldots, K_{\text{AnsAug}}\}$: the question is appended to a few in-context reasoning examples, then fed to the LLM for generating its reasoning path $r_i^{(j)}$ and answer $a_i^{(j)}$. We filter out reasoning paths with correct answers as:

$$\mathcal{D}_{\text{AnsAug}} = \{(q_i, r_i^{(j)}, a_i^{(j)}) : a_i^{(j)} = a_i^\star; i = 1, \ldots, N_q; j = 1, \ldots, K_{\text{AnsAug}}\}. \tag{1}$$

## 3.2 QUESTION BOOTSTRAPPING BY LLM REPHRASING

Generating more answers for mathematical questions with LLMs is straightforward, but creating questions is more challenging. Math Questions are written by well-educated teachers. Hence, enlarging the question set through manual creation is time-consuming and labor-intensive. To address this issue, we propose rephrasing prompting to generate more questions through the LLM.

> **Example 3.1: Rephrasing Question**
>
> **Question:** What is the total amount that James paid when he purchased 5 packs of beef, each weighing 4 pounds, at a price of \$5.50 per pound?
> **Answer:** Each pack of beef weighs 4 pounds, so 5 packs weigh 4 * 5 = 20 pounds in total. The price per pound of beef is \$5.50, so the total cost for 20 pounds is 20 * \$5.50 = \$110.  ...  The answer is: 110.

Specifically, for a question $q_i$, we append it to the prompt, which is then fed to the LLM for generating the rephrased question. Example 3.1 shows a generated rephrased question and the complete prompt is shown in Appendix A.1. We adopt temperature sampling to sample $K_{\text{rephrase}}$ rephrased questions for each meta-question. For the rephrased questions, it is time-consuming to manually check the consistency compared with the original questions. We propose a supervised method to evaluate the correctness between the rephrased questions and the meta-questions. For each rephrased question $\hat{q}_i^{(j)}$, we use few-shot Chain-of-Thought prompting to generate its reasoning path $\hat{r}_i^{(j)}$ and answer $\hat{a}_i^{(j)}$, which is compared with the ground-truth answer $a_i^\star$. The accuracy of Complexity-based CoT [20] for answering the rephrased question by GPT-3.5-Turbo is 76.30%, which is comparable to that of answering the original training questions (80.74%). This suggests that the quality of rephrased questions is preserved high while the question diversity is improved. We collect the rephrased questions with correct answers (*i.e.*, $\hat{a}_i^{(j)} = a_i^\star$) as the augmented data:

$$\mathcal{D}_{\text{rephrase}} = \{(\hat{q}_i, \hat{r}_i^{(j)}, \hat{a}_i^{(j)}) : \hat{a}_i^{(j)} = a_i^\star; i = 1, \ldots, N_q; j = 1, \ldots, K_{\text{rephrase}}\}. \tag{2}$$

## 3.3 QUESTION BOOTSTRAPPING BY BACKWARD REASONING

Backward reasoning plays an important role in answering many mathematical questions, *i.e.*, starting with a given condition and thinking backward to determine an unknown variable in the question. One specific example between a question and a backward question is illustrated in Example 3.2. However, existing methods (SFT, RFT, WizardMath) have significantly lower accuracy on backward questions, as shown in Figure 6, motivating us to bootstrap backward questions to improve the reasoning ability.

> **Example 3.2: Question and Backward Question**
>
> **Question:** James buys 5 packs of beef that are 4 pounds each. The price of beef is \$5.50 per pound. How much did he pay? **Answer:** He bought 5*4=20 pounds of beef. He paid 20*5.5=\$110. The answer is: 110 ✓
> **Backward Question:** James buys x packs of beef that are 4 pounds each. The price of beef is \$5.50 per pound. How much did he pay? If we know the answer to the above question is 110, what is the value of unknown variable x? **Answer:** The total weight of the beef is 4*x because 4*5.5 = 22. ... The answer is: 27 ✗

To improve the backward reasoning ability of finetuned models, we generate more questions which can be solved in a backward manner: a number in the question $q_i$ is masked by "x", while the LLM is

asked to predict the value of "x" when its answer $a_i^\star$ is provided. Different from forward reasoning, which generates explicit intermediate steps towards the final answer, backward reasoning starts with the answer and generates multiple reasoning steps to predict the masked number. Representative backward reasoning methods include Self-Verification [76] and FOBAR [32].

In Self-Verification (SV) [76], the question with the answer is first rewritten into a declarative statement, *e.g.*, "How much did he pay?" (with the answer 110) is rewritten into "He paid $10". Then, a question for asking the value of **x** is appended, *e.g.*, "What is the value of unknown variable **x**?". Example 3.3 gives an augmented example. We collect the new questions and their generated reasoning paths with correct answers as the augmented data:

$$\mathcal{D}_{\text{SV}} = \{(\tilde{q}_i^{(j)}, \tilde{r}_i^{(j)}, \tilde{a}_i^{(j)}) : \tilde{a}_i^{(j)} = a_i^\star; i = 1, \ldots, N_q; j = 1, \ldots, K_{\text{SV}}\}. \tag{3}$$

---

**Example 3.3: Self-Verification [76] Question**

**Question:** James buys x packs of beef that are 4 pounds each. The price of beef is $5.50 per pound. He paid 110. What is the value of unknown variable x?
**Answer:** To solve this problem, we need to determine the value of x, which represents the number of packs of beef that James bought. Each pack of beef weighs 4 pounds and ... The value of x is 5.

---

**Example 3.4: FOBAR [32] Question**

**Question:** James buys x packs of beef that are 4 pounds each. The price of beef is $5.50 per pound. How much did he pay? If we know the answer to the above question is 110, what is the value of unknown variable x?
**Answer:** James buys x packs of beef that are 4 pounds each, so he buys a total of 4x pounds of beef. The price of beef is $5.50 per pound, so the total cost of the beef is 5.50 * 4x = 22x. ... The value of x is 5.

---

Self-Verification needs to rewrite the question with an answer into a declarative statement, which is challenging for complex questions. To address this issue, FOBAR [32] proposes to directly append the answer to the question, *i.e.*, "*If we know the answer to the above question is* $\{a_i^\star\}$ *, what is the value of unknown variable* **x***?*" Example 3.4 shows an example. We collect the new questions along with their correct answers as our augmented data:

$$\mathcal{D}_{\text{FOBAR}} = \{(\bar{q}_i^{(j)}, \bar{r}_i^{(j)}, \bar{a}_i^{(j)}) : \bar{a}_i^{(j)} = a_i^\star; i = 1, \ldots, N_q; j = 1, \ldots, K_{\text{FOBAR}}\}. \tag{4}$$

### 3.4 FINETUNING OBJECTIVE FUNCTIONS

We merge all the augmented data, including answer-augmented data and bootstrapped questions (Rephrasing, Self-Verification, FOBAR) as $\mathcal{D}_{\text{MetaMathQA}} = \mathcal{D}_{\text{AnsAug}} \cup \mathcal{D}_{\text{rephrase}} \cup \mathcal{D}_{\text{SV}} \cup \mathcal{D}_{\text{FOBAR}}$. We finetune a LLM model (parameterized by $\boldsymbol{\theta}$) on $\mathcal{D}_{\text{MetaMathQA}}$ to obtain the MetaMath model by maximizing the log likelihood of the reasoning path conditioned on the question, *i.e.*, $\mathcal{L}(\boldsymbol{\theta}) = \sum_{(q,r,a) \in \mathcal{D}_{\text{MetaMathQA}}} \log \mathbb{P}(r \mid q; \boldsymbol{\theta})$. Although we only consider LLaMA-2 here, MetaMathQA can also be used to finetune other LLMs.

## 4 EXPERIMENTS AND RESULTS

### 4.1 EXPERIMENTAL SETUP

**Datasets.** We use two popular mathematical reasoning benchmarks: (i) GSM8K [13] is a dataset consisting of high-quality grade school math problems, containing 7,473 training samples and 1,319 testing samples; and

| Dataset | AnsAug | Rephrasing | SV | FOBAR | Overall |
|---|---|---|---|---|---|
| MetaMathQA-GSM8K | 80K | 80K | 40K | 40K | 240K |
| MetaMathQA-MATH | 75K | 50K | 15K | 15K | 155K |
| MetaMathQA | 155K | 130K | 55K | 55K | 395K |

Table 1: Number of samples in the proposed MetaMathQA.

(ii) MATH [23] dataset consists of high school math competition problems that span seven subjects including Prealgebra, Algebra, Number Theory, Counting and Probability, Geometry, Intermediate Algebra, and Precalculus. It contains 7,500 and 5,000 samples for training and testing, respectively. Questions in GSM8K [13] take between 2 and 8 steps to reach the answer, while MATH is much more challenging.

**Models.** We use the current state-of-the-art open-source model LLaMA-2 [70], including three different parameter sizes: 7B, 13B, and 70B, as the base model for fine-tuning. GPT-3.5-Turbo is used for rephrasing questions as well as generating answers in all four augmentations, where the temperature is set to 0.7 as in [74]. The LLaMA-2-7B and LLaMA-2-13B are trained by fully fine-tuning. LLaMA-2-70B is finetuned by QLoRA [15] for computational efficiency. More experimental details can be seen in Appendix B.

**Baselines.** The proposed methods are compared with (i) closed-source models such as GPT-3.5-Turbo [53], PaLM [12]; (ii) open-source models such as LLaMA-1 [69], LLaMA-2 [70]; (iii) Supervised Fine-Tuning (SFT), which uses the training set of the original GSM8K or MATH datasets; (iv) Rejection sampling Fine-Tuning (RFT) [79] generates and collects correct reasoning paths as augmented data for fine-tuning; (v) Wizard-Math [43] which generates samples and trains two reward models using ChatGPT [1] to select samples for fine-tuning.

**Diversity Gain.** We use the diversity gain [6] to measure to what extent a new dataset added to a basic dataset can improve the overall data diversity. For a base dataset $\mathcal{D}_{\text{base}} = \{x_i = (q_i, r_i, a_i)\}_{i=1}^{N}$ with $N$ samples, and a new dataset $\mathcal{D}_{\text{new}} = \{x_i = (q_i, r_i, a_i)\}_{i=1}^{M}$ with M samples, the diversity gain is defined as: $\mathcal{D}_{\text{new}}$ relative to $\mathcal{D}_{\text{base}}$ as: $d_{\text{gain}} = \frac{1}{M} \sum_{x_i \in \mathcal{D}_{\text{new}}} \min_{x_j \in \mathcal{D}_{\text{base}}} (\|f(x_i) - f(x_j)\|_2^2)$, where $f$ is the feature extractor and we use the OpenAI Embedding API *text-embedding-ada-002* for feature extraction. For Figure 2, we change the data size of base data and select a fixed set of 20K new data points that the model has not encountered to form $\mathcal{D}_{\text{new}}$.

## 4.2 RESULTS
ON GSM8K AND MATH

Table 1 illustrates the detailed description of our MetaMathQA collection and Table 2 shows the testing accuracy on GSM8K and MATH. As can be seen, for open-source models with 1-10B parameters, MetaMath achieves the state-of-the-art performance. Compared to the previous best LLM, MetaMath achieves a large

| Model | #params | GSM8K | MATH |
|---|---|---|---|
| *closed-source models* | | | |
| GPT-4 [54] | - | 92.0 | 42.5 |
| GPT-3.5-Turbo [53] | - | 80.8 | 34.1 |
| PaLM [12] | 8B | 4.1 | 1.5 |
| PaLM [12] | 62B | 33.0 | 4.4 |
| PaLM [12] | 540B | 56.5 | 8.8 |
| PaLM-2 [2] | 540B | 80.7 | 34.3 |
| Flan-PaLM 2 [2] | 540B | 84.7 | 33.2 |
| Minerva [35] | 8B | 16.2 | 14.1 |
| Minerva [35] | 62B | 52.4 | 27.6 |
| Minerva [35] | 540B | 58.8 | 33.6 |
| *open-source models (1-10B)* | | | |
| LLaMA-2 [70] | 7B | 14.6 | 2.5 |
| MPT [49] | 7B | 6.8 | 3.0 |
| Falcon [57] | 7B | 6.8 | 2.3 |
| Code-LLaMA [61] | 7B | 25.2 | 13.0 |
| InternLM [29] | 7B | 31.2 | - |
| GPT-J [71] | 6B | 34.9 | - |
| ChatGLM 2 [81] | 6B | 32.4 | - |
| Qwen [1] | 7B | 51.6 | - |
| Baichuan-2 [4] | 7B | 24.5 | 5.6 |
| SFT [70] | 7B | 41.6 | - |
| RFT [79] | 7B | 50.3 | - |
| MAmooTH-CoT [80] | 7B | 50.5 | 10.4 |
| WizardMath [43] | 7B | 54.9 | 10.7 |
| MetaMath | 7B | **66.5** | **19.8** |
| *open-source models (11-50B)* | | | |
| LLaMA-2 [70] | 13B | 28.7 | 3.9 |
| LLaMA-2 [70] | 34B | 42.2 | 6.2 |
| MPT [49] | 30B | 15.2 | 3.1 |
| Falcon [57] | 40B | 19.6 | 2.5 |
| GAL [68] | 30B | - | 12.7 |
| Platypus [50] | 13B | 25.7 | 2.5 |
| Orca-Platypus [50] | 13B | 38.4 | 3.0 |
| Vicuna [11] | 13B | 27.6 | - |
| Code-LLaMA [61] | 13B | 36.1 | 16.4 |
| Baichuan-2 [4] | 13B | 52.8 | 10.1 |
| SFT [70] | 13B | 50.0 | - |
| RFT [79] | 13B | 54.8 | - |
| MAmooTH-CoT [80] | 13B | 56.3 | 12.9 |
| WizardMath [43] | 13B | 63.9 | 14.0 |
| MetaMath | 13B | **72.3** | **22.4** |
| *open-source models (51-70B)* | | | |
| LLaMA-2 [70] | 70B | 56.8 | 13.5 |
| RFT [79] | 70B | 64.8 | - |
| Platypus [50] | 70B | 70.6 | 15.6 |
| MAmooTH-CoT [80] | 70B | 72.4 | 21.1 |
| WizardMath [43] | 70B | 81.6 | 22.7 |
| MetaMath[‡] | 70B | **82.3** | **26.6** |

Table 2: Comparison of testing accuracy to existing LLMs on GSM8K and MATH. [‡]Due to the computing resource limitation, we finetune MetaMath-70B using QLoRA [15].

---

[1] https://openai.com/

| Method | GSM8K | | | | | | MATH | | | | | |
| --- | --- | --- | --- | --- | --- | --- | --- | --- | --- | --- | --- | --- |
| | AnsAug | Rep. | SV | FOBAR | GSM8K | MATH | AnsAug | Rep. | SV | FOBAR | GSM8K | MATH |
| SFT [70] | ✗ | ✗ | ✗ | ✗ | 41.6 | 3.0 | ✗ | ✗ | ✗ | ✗ | 13.8 | 4.7 |
| MetaMath | ✓ | ✗ | ✗ | ✗ | 59.6 | 4.4 | ✓ | ✗ | ✗ | ✗ | 28.4 | 12.9 |
| | ✗ | ✓ | ✗ | ✗ | 59.7 | 4.4 | ✗ | ✓ | ✗ | ✗ | 30.4 | 12.4 |
| | ✓ | ✓ | ✗ | ✗ | 60.6 | 4.4 | ✓ | ✓ | ✗ | ✗ | 29.1 | 15.3 |
| | ✓ | ✓ | ✓ | ✓ | **64.4** | **5.7** | ✓ | ✓ | ✓ | ✓ | **34.6** | **17.7** |

Table 3: Effect of different question augmentation with LLaMA-2-7B finetuned on GSM8K or MATH.

improvement of 11.6% on GSM8K and 9.1% on MATH in testing accuracy, showing that finetuning on our MetaMathQA data is effective.

As for LLMs with 11-50B parameters, the proposed MetaMath performs the best. Particularly, on both GSM8K and MATH, MetaMath achieves higher accuracy than SFT, RFT, and WizardMath by a large margin (+7%), demonstrating the effectiveness of the MetaMath data in improving mathematical reasoning ability. Furthermore, for LLMs with 51-70B parameters, again, MetaMath achieves the highest testing accuracy. Particularly, MetaMath is better than GPT-3.5-Turbo on GSM8K, which is used for generating augmented data for finetuning.

## 4.3 EFFECT OF AUGMENTATIONS

In this section, we conduct experiments to study the effect of augmentations in MetaMath. We first finetune the LLaMA-2-7B model on augmented GSM8K (MetaMath-GSM8K) data, and test the finetuned model on GSM8K and MATH. Table 3 shows the testing accuracy of different combinations of augmentations, where we mix all augmented data together for each model. As can be seen, on GSM8K, the models trained on answer augmentation (AnsAug) or rephrasing augmentation achieve much higher accuracy than SFT, which is only trained on the training set. Combing answer augmentation and rephrasing augmentation data for fine-tuning leads to a slightly higher accuracy, which is further improved by about 4% through merging the FOBAR and SV augmentation data. As for MATH, MetaMath trained only on MetaMahQA-GSM8K data performs better than SFT, suggesting its effectiveness in generalizing to unseen mathematical tasks.

We also conduct an experiment by fine-tuning LLaMA-2-7B on the augmented MATH (MetaMathQA-MATH) data then evaluate the model on GSM8K and MATH. Table 3 shows the testing accuracy. Again, MetaMath trained on AnsAug or rephrasing augmentation data performs much better than SFT. Furthermore, merging all augmented data together for fine-tuning is better than merging AnsAug and rephrasing augmentation data, demonstrating the effectiveness of SV and FOBAR augmentation data in improving mathematical reasoning ability. Moreover, for the unseen GSM8K task, MetaMath trained on MetaMathQA-MATH data is significantly better than SFT (+20%).

## 4.4 DISCUSSION FROM A PERPLEXITY PERSPECTIVE

According to the Superficial Alignment Hypothesis proposed by Zhou et al. [83], the capability of a model is rooted in pretraining, and data from downstream tasks acts to activate the inherent ability of LLMs that has been learned during pretraining. There are two important questions that arise from such a hypothesis: (i) *what* kind of data is most effective at activating possible latent knowledge, and (ii) *why* is one dataset better than another at such activation? Our empirical results suggest that, in the mathematical tasks we consider, our MetaMathQA dataset may serve as a superior activator of mathematical knowledge. Yet, *why* MetaMath yields superior performance than training on the data of correct answer-only or GSM8K CoT is unclear. We speculate that perhaps it is

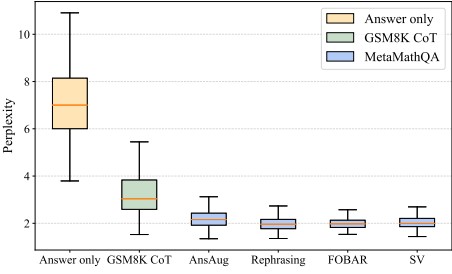

Figure 3: Lower perplexity of MetaMathQA.

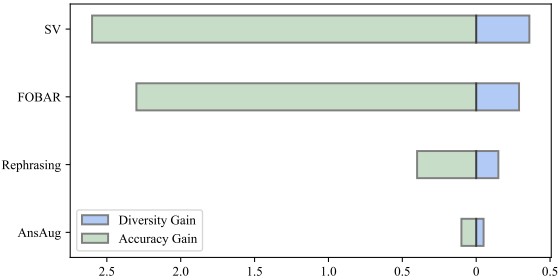

Figure 4: Accuracy correlates positively with diversity.

the simplicity of the data that matters. As shown in Figure 3, we compute the perplexity [46, 72] for the under-finetuned LLaMA-2-7B model, in terms of answer-only data, GSM8K CoT, and the subsections of MetaMathQA data. The perplexity of MetaMathQA is significantly lower than the other two datasets. This highlights its inherently easy-to-learn nature, which may be more conducive to eliciting bolstered problem-solving abilities from an LLM. This is also aligned with the findings with TinyStories [18], where short and easy story data can help LLMs generate content fluently.

## 4.5 DISCUSSION FROM A DIVERSITY PERSPECTIVE

As shown in Figure 2, naively prompting GPT-3.5-Turbo for answer augmentation leads to a clear accuracy saturation. After accuracy saturation, increasing the AnsAug data only yields a limited performance gain. For instance, using 80K answer augmentation data to train a LLaMA-2 7B model leads to a 59.6% accuracy, adding new 20K AnsAug data would only take 0.1% performance gain. This is due to the homogeneity of the additional samples, contributing to a diversity gain of only 0.05 (shown in Figure 4). In comparison, adding the same amount of data generated by question bootstrapping leads to a significant performance boost, which is due to the noticeable diversity gain brought by question bootstrapping. As shown in Figure 4, adding 20K data from Rephrasing, FOBAR, or SV takes an increasing diversity gain, thus causing a 0.4%, 2.3%, and 2.6% accuracy gain, respectively. This experiment demonstrates a positive correlation (the Pearson coefficient is 0.972) between the diversity brought by the bootstrapping methods and accuracy. This is also aligned with the success of MetaMath, which is trained with the diverse MetaMathQA dataset including 4 kinds of data reflecting both the forward and backward reasoning paths.

## 4.6 EVALUATING THE REVERSAL MATHEMATICAL CAPABILITY

The Reversal Curse [5], where LLMs trained from a sentence "A is B" are not able to generalize to answer "B is A", also aligns with the observation in this paper that LLMs lack backward mathematical reasoning ability. To evaluate the backward mathematical capability, we propose a GSM8K-Backward test set, including 1270 backward questions by using SV and FOBAR to augment the original GSM8K test set (as shown in Example 3.3 and Example 3.4). Figure 6 shows the accuracy comparison of different 7B mathematical LLMs between the GSM8K and GSM8K-Backward datasets. As can be seen, existing LLMs struggle to solve mathematical problems in backward rationales and our MetaMath has a significant improvement on both datasets. Specifically, the ways where different LLMs solve the backward mathematical problem are illustrated through examples in Appendix C.

## 4.7 REASONING PATHS WITH INCORRECT ANSWER CAN ALSO BE USEFUL

We conduct experiments on GSM8K using LLaMA-2-7B to study whether the answer augmentation samples with incorrect answers are helpful for finetuning the LLM. We randomly choose 7,473 reasoning paths with incorrect answers from the generated answers, and we ensure that the size is the same as that of the original training set. From Table 4, we observe that the model finetuned on the augmented data with incorrect answers is still better than SFT, which is counter-intuitive. We hypothesize that although the final answer is incorrect,

| Data | Accuracy |
|---|---|
| GSM8K [13] | 41.6 |
| Incorrect Answers | 43.6 |
| Correct Answers | **52.2** |

Table 4: Testing accuracy on GSM8K of LLaMA-2-7B trained on different data.

some intermediate reasoning steps are correct (see Example 4.1). These reasoning steps can still be useful supervision signals. Our results are also aligned with [39], where they discover the importance of intermediate process supervision for reasoning.

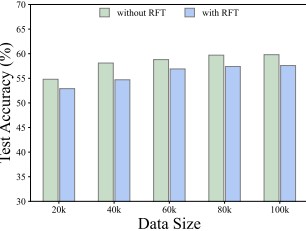

Figure 5: Combing RFT [79] dataset with our MetaMathQA leads to a performance drop.

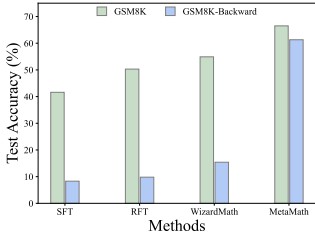

Figure 6: The accuracy gap between GSM8K and GSM8K-Backward.

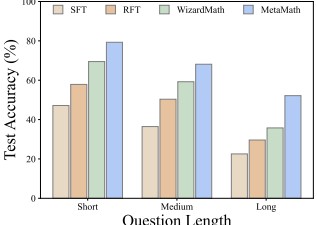

Figure 7: Testing accuracy on questions with short length, medium length and long length.

---

**Example 4.1: A Reasoning Path with Incorrect Answer**

**Question:** Tonya is in a hamburger eating contest. Each hamburger is 4 ounces. Last year the winner ate 84 ounces. How many hamburgers does she have to eat to beat last year's winner? (Ground-truth answer is 22)
**Generated Answer:** If each hamburger is 4 ounces and last year's winner ate 84 ounces, then Tonya needs to eat 84/4 = 21 hamburgers to beat last year's winner. The answer is: 21

---

## 4.8 MORE DATA IS NOT ALWAYS BETTER

There are also previous works that augment mathematical reasoning data for fine-tuning [43, 79]. An interesting question is whether combining existing augmented datasets with our MetaMathQA can improve the overall mathematical problem-solving performance. We select the RFT [79] dataset as the external dataset. Figure 5 shows that merging the RFT data into MetaMathQA actually hurts the performance, indicating that the RFT data may not be beneficial to MetaMath. Such a phenomenon is consistently observed in the MetaMathQA dataset under different sizes (from 20K to 100K), and the added RFT dataset is about 47K. The performance drop implies that more augmented data does not always help the generalization.

---

**Example 4.2: Case Study in Test Set of GSM8K**

**Question:** Darrell and Allen's ages are in the ratio of 7:11, If their total age now is 162, calculate Allen's age 10 years from now. (Ground-truth answer is 109)
**SFT Answer:** The total ratio representing their ages is 7+11=<<7+11=18>>18. The fraction representing Darrell's age is 7/18, and since their total age now is 162, Darrell's age is 7/18*162 = <<7/18*162=60>>60. The fraction representing Allen's age is 11/18, and since their total age now is 162, Allen's age is 11/18*162 = <<11/18*162=84>>84. Ten years from now, Allen's age will be 84 + 10 = <<84+10=94>>94.
**RFT Answer:** The total ratio representing Darrell and Allen's ages is 7+11 = <<7+11=18>>18. The fraction representing Darrell's age from the total ratio is 7/18, and since their total age is 162, Darrell is 7/18*162 = 100 years old. Allen's age, from the total ratio, is 11/18, and since Darrell is 100, Allen is 11/18*109 = <<11/18*100=66>>66 years old. Ten years from now, Allen will be 66 + 10 = 76 years old.
**WizardMath Answer:** We know Darrell and Allen's ages are in the ratio of 7:11, which means the total ratio representing their ages is 7+11 = 18. If their total age now is 162, we can calculate Darrell's age by using the ratio. Specifically, Darrell's age is 7/18 of 162, which is 7/18*162 = 49 years old. ... The answer is: 76.
**Our Answer:** The ratio of Darrell's age to Allen's age is 7:11. Let's assume Darrell's age is 7x and Allen's age is 11x. The total age of Darrell and Allen is 7x + 11x = 18x. We are given that the total age is 162, so 18x = 162. Dividing both sides by 18, we get x = 9. Therefore, Allen's age is 11x = 11 * 9 = 99. 10 years from now, Allen's age will be 99 + 10 = 109.

---

## 4.9 ERROR ANALYSIS

We have demonstrated that – across multiple scales – our MetaMath models can achieve stellar problem-solving performance. Yet, it is important to consider the characteristics of problems that induce errors in MetaMath and existing open-source mathematical models. In particular, we consider the relationship between question length and model performance. To investigate, we divide the GSM8K test set into three equally-sized subsets based on the different lengths of questions and calculate the accuracy of the models over each subset. We find in Figure 7 that, MetaMath and related methods struggle under longer questions. However, excitingly, MetaMath always obtains superior performance. We see the study of improving model performance with longer question lengths – for instance, by further augmenting the MetaMathQA dataset – as ripe grounds for future work.

## 5 CONCLUDING REMARKS

In this paper, we focus on improving the mathematical problem-solving abilities of open-source LLMs. By bootstrapping mathematical questions on GSM8K and MATH, we present a high-quality and diverse dataset MetaMathQA, involving forward reasoning and backward reasoning samples. Our family of LLMs finetuned on MetaMathQA, called MetaMath, have achieved state-of-the-art on mathematical benchmarks among all open-source LLMs. Remarkably, MetaMath-7B reaches $66.5\%$ on GSM8K and $19.8\%$ on MATH, surpassing previous open-source LLMs by a significant margin. Our work further emphasizes the importance of the characteristics of the training data on boosting LLM problem-solving capabilities.

ACKNOWLEDGEMENT

The authors would like to sincerely thank Katherine M. Collins from University of Cambridge for her valuable insights and suggestions.

This work was supported by NSFC key grant 62136005, NSFC general grant 62076118, and Shenzhen fundamental research program JCYJ20210324105000003. This research was supported in part by the Research Grants Council of the Hong Kong Special Administrative Region (Grants 16200021 and 16202523). AW acknowledges support from a Turing AI Fellowship under grant EP/V025279/1, and the Leverhulme Trust via CFI.

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

# Appendix

## Table of Contents

# A PROMPTS

## A.1 REPHRASING PROMPTS

---

**Example A.1: Prompt for Rephrasing GSM8K Questions**

*You are an AI assistant to help me rephrase questions. Follow the given examples.*

**Question:** Olivia has $23. She bought five bagels for $3 each. How much money does she have left?
**Rephrase the above question:** What is the amount of money that Olivia has left after purchasing five bagels for $3 each, if she initially had $23?

**Question:** Michael had 58 golf balls. On tuesday, he lost 23 golf balls. On wednesday, he lost 2 more. How many golf balls did he have at the end of wednesday?
**Rephrase the above question:** After losing 23 golf balls on Tuesday and an additional 2 on Wednesday, how many golf balls does Michael have left if he initially had 58 golf balls?

**Question:** Angelo and Melanie want to plan how many hours over the next week they should study together for their test next week. They have 2 chapters of their textbook to study and 4 worksheets to memorize. They figure out that they should dedicate 3 hours to each chapter of their textbook and 1.5 hours for each worksheet. If they plan to study no more than 4 hours each day, how many days should they plan to study total over the next week if they take a 10-minute break every hour, include 3 10-minute snack breaks each day, and 30 minutes for lunch each day?
**Rephrase the above question:** Angelo and Melanie need to study 2 chapters in their textbook and 4 worksheets for their upcoming test. They have planned to dedicate 3 hours for each chapter and 1.5 hours for each worksheet. They can study for a maximum of 4 hours each day, taking into account 10-minute breaks every hour, 3 10-minute snack breaks per day, and 30 minutes for lunch. How many days do they need to study in total over the next week to complete their study plan?

**Question:** Leah had 32 chocolates and her sister had 42. If they ate 35, how many pieces do they have left in total?
**Rephrase the above question:** If Leah had 32 chocolates and her sister had 42, and they both consumed 35 chocolates, what is the total number of chocolates that they have left?

**Question:** There were nine computers in the server room. Five more computers were installed each day, from monday to thursday. How many computers are now in the server room?
**Rephrase the above question:** If there were initially nine computers in the server room and five more computers were added each day from Monday to Thursday, what is the current total number of computers in the server room?

**Question:** Jason had 20 lollipops. He gave Denny some lollipops. Now Jason has 12 lollipops. How many lollipops did Jason give to Denny?
**Rephrase the above question:** If Jason initially had 20 lollipops and now has 12 after giving some to Denny, how many lollipops did he give to Denny?

**Question:** Sam bought a dozen boxes, each with 30 highlighter pens inside, for $10 each box. He rearranged five of these boxes into packages of six highlighters each and sold them for $3 per package. He sold the rest of the highlighters separately at the rate of three pens for $2. How much profit did he make in total, in dollars?
**Rephrase the above question:** Sam purchased 12 boxes, each containing 30 highlighter pens, at $10 per box. He repackaged five of these boxes into sets of six highlighters and sold them for $3 per set. He sold the remaining highlighters individually at a rate of three pens for $2. What is the total profit he made in dollars?

**Question:** There are 15 trees in the grove. Grove workers will plant trees in the grove today. After they are done, there will be 21 trees. How many trees did the grove workers plant today?
**Rephrase the above question:** If there were initially 15 trees in the grove and the grove workers are planning to plant more trees today, resulting in a total of 21 trees, how many trees did the workers plant today?

**Question:** {Q}
**Rephrase the above question:**

---

## A.2 Rewriting Question with Answer into a Declarative Statement

> **Example A.2: Prompts for Rewriting Question with Answer into a Declarative Statement**
>
> *You are an AI assistant to help me rewrite question into a declarative statement when its answer is provided. Follow the given examples and rewrite the question.*
>
> **Question:** How many cars are in the parking lot? The answer is: 5.
> **Result:** There are 5 cars in the parking lot.
> ...
> **Question:** {Q} The answer is: {A}.
> **Result:**

## B Experimental Details

**Training Details.** For the fully fine-tuning setting, we use the AdamW optimizer to train the model with 3 epochs and the batch size is 128. We use 8 NVIDIA A100 GPUs to train the 7B and 13B models, the learning rate is set as 2e-5 with a 3% learning rate warmup. For the 70B model QLoRA fine-tuning, the LoRA rank and alpha are 96 and 16, with a 0.05 dropout between the two matrices. The LoRA matrices are append in both the attention layer and the mlp layer. We use the same AdamW optimizer but with a 1e-4 learning rate and without a learning rate warmup. The Training Prompt 1 are basically from Alpaca [67], where the instruction is replaced by the MetaMathQA question.

> **Prompt 1: Training Prompt**
>
> *Below is an instruction that describes a task. Write a response that appropriately completes the request.\n\n### Instruction:\n{instruction}\n\n### Response:*

> **Prompt 2: Evaluation Prompt**
>
> *Below is an instruction that describes a task. Write a response that appropriately completes the request.\n\n### Instruction:\n{instruction}\n\n### Response: Let's think step by step.*

**Evaluation Prompting.** Different from the few-shot prompting evaluation for closed-source models, we find that zero-shot prompting is better for finetuned LLMs, which also saves more inference costs. Hence, MetaMath uses the zero-shot Evaluation Prompt 2 for GSM8K and MATH, where the instruction is replaced by the testing question. We set the temperature as 0 for fine-tuned LLaMA model.

**Answer Extraction.** Different from the Wei et al. [75], where they use complex string rules to extract the final answer. In line with WizardMath [43], MetaMath only extracts the string behind *The answer is:* as the final answer. To teach the model this extraction method, we append *The answer is:* {*gold answer*} to the end of answers in the MetaMathQA dataset, where the gold answer is replaced by the respective question's answer.

## C HOW DO DIFFERENT LLMS SOLVE REVERSAL MATHEMATICAL PROBLEMS?

---

**Example C.1: Case study about Reverse Mathematical problem**

**Question:** A robe takes 2 bolts of blue fiber and x% that much white fiber. It takes a total of 3 bolts. What is the value of unknown variable x? (Ground-truth answer is 50)

**SFT Answer:** We know that the robe takes 2 * x% = 3 bolts. So 2 * x% = 3. So x% = 3/2 = 60%. So x = 60. #### 60. The answer is: 60

**RFT Answer:** We know that the robe takes 2 bolts of blue fiber and x% that much white fiber, so it takes 2 * 0.01 = 0.02 bolts of white fiber. So in total, it takes 2 + 0.02 = ¡¡2+0.02=2.02¿¿2.02 bolts of blue and white fiber. Since we know that it takes 3 bolts in total, we can write the equation 2.02 = 3. We can solve for x by multiplying both sides by 3, getting 6.04 = 3 * 0.02. We can then write the equation 6.04 = 3, and solve for x by dividing both sides by 6.04, getting x = 0.02. #### 2 The answer is: 2

**WizardMath Answer:** Step 1: Define the variables. Let x% of the blue fiber be represented by the variable y. Then, the robe takes 2y bolts of blue fiber. Step 2: Set up the equation. Since the robe takes a total of 3 bolts, we can write the equation as follows: 2y + xy = 3 Step 3: Simplify the equation. We can simplify the equation by combining the terms with y: 2y + xy = 3 3y + xy = 3 Step 4: Solve for x. To solve for x, we need to isolate xy on one side of the equation. We can do this by subtracting 3y from both sides: 3y + xy - 3y = 3 - 3y xy = 3 - 3y Step 5: Find the value of x. Now, we need to find the value of x. We know that xy = 3 - 3y, and we want to express y in terms of x. To do this, we can divide both sides of the equation by x: (3 - 3y) / x = y Step 6: Solve for y. Now, we can solve for y by multiplying both sides of the equation by x: 3 - 3y = xy 3 - 3y = x(3 - 3y) Step 7: Simplify the equation. We can simplify the equation by expanding and combining like terms: 3 - 3y = 3 - 3y Step 8: Find the value of x. Now, we can see that y = 0, which means that x% of the blue fiber is 0%. Since we are dealing with a percentage, the only possible value for x is 0. Therefore, the value of unknown variable x is 0. Final Answer: x = 0. The answer is: x

**Our Answer:** The robe takes 2 bolts of blue fiber. It takes x% of that much white fiber, which means it takes (x/100) * 2 bolts of white fiber. In total, it takes 3 bolts. So, we can write the equation as: 2 + (x/100) * 2 = 3 Let's simplify and solve for x: 2 + (2x/100) = 3 To isolate x, we subtract 2 from both sides of the equation: 2 + (2x/100) - 2 = 3 - 2 (2x/100) = 1 To solve for x, we multiply both sides of the equation by 100/2: (2x/100) * (100/2) = 1 * (100/2) x = 50 The value of x is 50. #### 50 The answer is: 50

---

## D GENERALIZATION TO GAME OF 24

The core idea of the proposed bootstrapping approach is to diversifying the questions in both forward and backward reasoning directions, where our approach can be extended to other reasoning tasks easily. We conducted an additional experiment to show a successful application of our bootstratpping method to the Game of 24, which involves multi-hop reasoning steps to attain 24 given 4 numbers. Given an original question with 4 numbers (2,3,4,12), its answer (2*3-4)*12 is an mathematical expression that includes the 4 numbers and uses basic arithmetic operations (+, -, *, /) to reach 24. In Game of 24, We can also apply answer augmentation and question bootstrapping to generate more question-answer pairs to diversify the training data. The details of answer augmentation and question bootstrapping for Game of 24 is as following:

**Answer Augmentation.** The solutions of obtaining 24 given 4 numbers may not be unique, e.g., (23-4)12 = 24 and 212(4-3) = 24 are two different solutions for the given numbers (2,3,4,12). In Answer Augmentation, we enumerate all the correct solutions for the given question with 4 numbers and collect all the solutions as the Answer Augmentation data, which exactly matches the core idea of Answer Augmentation in GSM8K & MATH: Augment data by diversifying the paths of answers without altering the question.

**Question Bootstrapping.** Game of 24 can be extended to **Game of** $n$, i.e., given 4 numbers (one number is 24), the goal is to obtain $n$ using basic arithmetic operations (+, -, , /). We use Game of n for question bootstrapping. We replace a number in the original question with 24 and the question is to obtain the substituted number. This idea is similar to create backward questions in our paper, i.e., masking a number in the question and asking the LLM to predict the number. For a Game of 24 question, we can bootstrap it and obtain 4 Game of $n$ questions, as an example show in Table 5.

**Game of 24 Setup.** We randomly select 1362 Game of 24 questions from www.4nums.com, where 681 questions are for training and the remaining 681 questions are held-out for testing. We apply the above augmentation methods to generate more training data from the 681 questions: (i) apply answer augmentation by enumerating all the correct forward solutions and obtain an AnsAug datasets consists of 6052 question-answer pairs; (ii) apply question bootstrapping to obtain bootstrapping dataset

|  | Bootstrapping1 | Bootstrapping2 | Bootstrapping3 | Bootstrapping4 |
|---|---|---|---|---|
| Input (4 numbers) | 24, 3, 4, 12 | 2, 24, 4, 12 | 2, 3, 24, 12 | 2, 3, 4, 24 |
| Target (n) | 2 | 3 | 4 | 12 |
| Solution | (4-3)/(12/24) = 2 | (24/12+4)/2 = 3 | 24/12*3-2 = 4 | (24/4-2)*3 = 12 |

Table 5: Illustration of question bootstrapping: from Game of 24 to Game of $n$.

| Method | #Samples | Accuracy |
|---|---|---|
| SFT | 681 | 1.8 |
| AnsAug | 6052 | 10.2 |
| AnsAug + Bootstrapping | 6052 | **12.0** |

Table 6: Accuracy comparison on Game of 24 between our bootstrapping method and ansaug.

(consists of 2724 Game of n question-answer pairs). To verify the effectiveness of the bootstrapping approach, we randomly sample 4000 question-answer pairs (Game of 24) from the AnsAug datasets, and 2052 backward question-answer pairs (Game of n) from the bootstrapping dataset. We finetune LLaMA-2-7B on AnsAug data and the mixed data separately for comparison.

**Results on Game of 24.** Table 6 shows the testing accuracy. As can be seen, our proposed augmentation approaches (AnsAug and AnsAug+Bootstrapping) have higher accuracy than SFT, which trains on the original 681 question-answer pairs. Furthermore, using question bootstrapping for augmentation can boost the performance of AnsAug. Hence, the proposed bootstrapping method is also effective for other multi-hop reasoning tasks, such as Game of 24.

**Results on Game of n.** For each question-answer pair in the testing set of **Game of 24**, we create 4 more testing questions of **Game of n** using the above question boostrapping method. In total, we obtain 3405 testing questions. Table 7 shows the testing accuracy. Again, using our augmentation methods (both AnsAug and Bootstrapping) perform better than SFT by a large margin. Furthermore, AnsAug + Bootstrapping performs the best, demonstrating our proposed method is also useful for Game of n.

| Method | #Samples | Accuracy |
|---|---|---|
| SFT | 681 | 0.8 |
| AnsAug | 6052 | 3.0 |
| AnsAug + Bootstrapping | 6052 | **8.1** |

Table 7: Accuracy comparison on Game of $n$ between our bootstrapping method and ansaug.

# E    MORE EXPERIMENTAL RESULTS

## E.1    METAMATHQA IS USEFUL FOR DIFFERENT BASE MODELS

We conduct additional experiments to verify the generalizability of the MetaMathQA dataset across different base models. In addition to LLaMA-2-7B and LLaMA-2-13B, We finetune two more powerful base models Mistral-7B [30] and Llemma-7B [3] on MetaMathQA. Table 8 shows the testing accuracy on GSM8K and MATH. As can be seen, our proposed MetaMathQA is consistently useful for all four base models. Moreover, the improvements brought by MetaMathQA are large.

| Base Model | MetaMathQA | GSM8K | MATH |
|---|---|---|---|
| LLaMA-2-7B [70] | ✗ | 14.6 | 2.5 |
|  | ✓ | **66.5** | **19.8** |
| LLaMA-2-13B [70] | ✗ | 28.7 | 3.9 |
|  | ✓ | **72.3** | **22.4** |
| Llemma-7B [3] | ✗ | 36.4 | 18.0 |
|  | ✓ | **69.2** | **30.0** |
| Mistral-7B [30] | ✗ | 52.2 | 13.1 |
|  | ✓ | **77.7** | **28.2** |

Table 8: Effectiveness of MetaMathQA on different base models.

## E.2    TESTING ACCURACY UNDER DIFFERENT AUGMENTATION DATA SIZE

In Figure 2, we have shown the proposed question bootstrapping method can boost the testing accuracy by a large margin, while the AnsAug method would quickly reach a state of saturation. We increase the AnsAug data to 240K and compare the performance of LLaMA-2-7B finetuned on AnsAug data (i.e., w/o Question Bootstrapping) and MetaMathQA-GSM8K with question bootstrapping (i.e., w/ Question Bootstrapping). We also conduct additional experiments on a larger model LLaMA-2-13B and Mistral-7B with a different architecture. Figures 8, 9, and 10 show the trends using LLaMA-2-7B, LLaMA-2-13B, and Mistral-7B, respectively. For all three models, we can see that finetuning on AnsAug rapidly reaches a state of accuracy saturation and continually increasing AnsAug data is hard to boost performance. In contrast, the test accuracy, when using bootstrapped questions, continues to exhibit a steady increase when AnsAug quickly saturates.

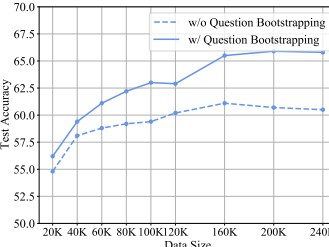
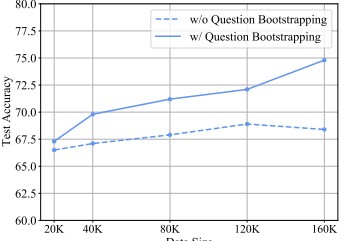

Figure 8: LLaMA-2-7B.          Figure 9: LLaMA-2-13B.          Figure 10: Mistral-7B.

### E.3   ABLATION STUDY ON A LARGER MODEL LLAMA-2-13B

In addition to the ablation study on LLaMA-2-7B (Table 3), we conducted an addition experiment to study the effect of augmentations in MetaMath using a larger model LLaMA-2-13B. Table 9 shows the testing accuracy. We can see that the observations are consistent with that of LLaMA-2-7B in Section 4.3: (i) Combing answer augmentation and rephrasing augmentation data for fine-tuning leads to a slightly higher accuracy. (ii) The accuracy can be further improved by merging the FOBAR and SV augmentation data.

| Method | AnsAug | Rep. | SV | FOBAR | GSM8K | MATH |
|---|---|---|---|---|---|---|
| SFT [70] | ✗ | ✗ | ✗ | ✗ | 50.9 | 4.5 |
| MetaMath | ✓ | ✗ | ✗ | ✗ | 66.0 | 5.5 |
| | ✗ | ✓ | ✗ | ✗ | 67.5 | 5.9 |
| | ✓ | ✓ | ✗ | ✗ | 68.1 | 5.8 |
| | ✓ | ✓ | ✓ | ✓ | **72.3** | **7.2** |

Table 9: Effect of different question augmentations with LLaMA-2-13B finetuned on GSM8K.

### E.4   OUT-OF-DISTRIBUTION ABILITY

| | #Params | Accuracy (Exact Match) |
|---|---|---|
| SFT | 7B | 25.8 |
| RFT | 7B | 26.7 |
| WizardMath | 7B | 31.5 |
| MetaMath | 7B | **37.1** |
| WizardMath | 13B | 46.4 |
| MetaMath | 13B | **49.5** |
| WizardMath | 70B | 63.1 |
| MetaMath | 70B | **72.3** |

Table 10: Exact Match Accuracy on DROP using zero-shot evaluation.

To investigate Out-of-Distribution ability of different models, we perform zero-shot evaluation on DROP [17] to compare MetaMath with baseline models. Since all these models targets at mathematical reasoning, we only consider the DROP questions with numerical answers. Table 10 shows the testing accuracy. As can be seen, MetaMath-7B and MetaMath-13B still outperform the baseline models by a large margin, demonstrating MetaMath does not suffer a benchmark hacking on GSM8K and MATH.

