# OpenReview forum: "MetaMath: Bootstrap Your Own Mathematical Questions for Large Language Models"
_ICLR.cc/2024/Conference — ICLR 2024 spotlight_

### Official Review · Reviewer_mqn7 · 2023-10-31

**Soundness:** 4 excellent
**Presentation:** 4 excellent
**Contribution:** 3 good
**Rating:** 8
**Confidence:** 4

**Summary:**

The authors aim to bridge the noticeable performance gap of open-access LLMs in solving complex mathematical problems. The paper introduces a framework that includes (i) a diverse dataset of math problems generated through transformations such as forward-backward reasoning and self-verification (MetaMathQA) and (ii) open-access LLMs (llama series) fine-tuned on MetaMathQA. Experiments on benchmark datasets demonstrate clear and impressive gains with MetaMath over other open LLMs. Additionally, the authors conduct insightful analyses, highlighting the role of question diversity in enhancing LLM performance.

**Strengths:**

- **Novel Approach:** The paper introduces a unique data augmentation strategy for mathematical reasoning. The MetaMath framework is generic and can be easily extended to other numerical reasoning datasets.

- **Rich and Comprehensive Analysis:** The analysis is rich and comprehensive, offering numerous insights into data augmentation and the fine-tuning of LLMs for reasoning tasks.

**Weaknesses:**

- **Potential for Benchmark Hacking:** Given the experimental setup, there is a slight risk that the proposed approach could lead to benchmark hacking.

- **Dependence on High-Quality Initial Questions:** Given that both datasets used have extensive training data available, the performance of the proposed method in the absence of high-quality initial questions available for mutation remains uncertain.

To some extent, both the weaknesses can be addressed by doing 0-shot evaluation on some other datasets like DROP (https://allenai.org/data/drop)

**Questions:**

In Table 3, MetaMath finetuning always begins with the AnsAug split, right? Do the authors have any thoughts on what would happen if we start training from (say) SV or FOBAR?

---

> ### Author Response · Authors · 2023-11-18
> **Response to Reviewer mqn7**
>
> We would like to sincerely thank the reviewer for the useful comments on our work. We take every comment seriously and hope our response can address the reviewer’s concerns. If there are any remaining questions, we are more than happy to address them.
>
> > Q1. **Potential for Benchmark Hacking & Dependence on High-Quality Initial Questions. To some extent, both the weaknesses can be addressed by doing 0-shot evaluation on some other datasets like DROP (https://allenai.org/data/drop)**
>
> **A1.** As suggested, we perform a zero-shot evaluation on the DROP dataset to compare MetaMath with baseline models. Since we focus on mathematical reasoning, we only consider the DROP questions with numerical answers. The table below shows the testing accuracy. As can be seen, MetaMath-7B and MetaMath-13B still outperform the baseline models by a large margin, which shows MetaMath does not suffer benchmark hacking on GSM8K and MATH datasets. We have added the results in the updated paper (Table 10 in Appendix A.8).
>
> | Method | #params | Accuracy (Exact Match) |
> |  :---- |  :----: | :----: |
> |SFT | 7B | 25.8 |
> |RFT| 7B | 26.7 |
> |WizardMath| 7B | 31.5|
> |MetaMath|7B| **37.1**|
> |||
> |WizardMath|13B| 46.4|
> | MetaMath |13B| **49.5** |
>
> ---
>
>
> > Q2. **In Table 3, MetaMath finetuning always begins with the AnsAug split, right? Do the authors have any thoughts on what would happen if we start training from (say) SV or FOBAR?**
>
> **A2.** We apologize for the confusion caused by Table 3. When using data from multiple bootstrapping methods for finetuning, we mix all augmented data instead of using them sequentially (AnsAug -> Rephrasing -> SV -> FOBAR). In other words, at each training step, we randomly take a batch of samples from the mixed augmented data for updating the model parameters.

---

> ### Comment · Reviewer_mqn7 · 2023-11-21
> **Thanks for your response**
>
> Thanks for including the results on DROP and for the clarification. The gains are impressive and convincing. Did you get a chance to try the 70B model? I understand if running inference on 70b is not possible due to resource constraints.

---

> > ### Author Response · Authors · 2023-11-23
> > **Response to further comments**
> >
> > Thanks for your further comments.
> >
> > > Q3. Did you get a chance to try the 70B model? I understand if running inference on 70b is not possible due to resource constraints.
> >
> > **A3.** As suggested, we perform an evaluation on the DROP dataset to compare MetaMath with WizardMath using LLaMA-2 (70B) as the base model. The table below shows the accuracy. As shown, MetaMath-70B achieves better performance.
> >
> > | Method | #params | Accuracy(Exact Match) |
> > |  :---- |  :----: | :----: |
> > |WizardMath| 70B |  63.1 |
> > |MetaMath|70B|  **72.3** |
> >
> > Thank you again for your efforts and time in improving our work.

---

### Official Review · Reviewer_AmB6 · 2023-11-01

**Soundness:** 4 excellent
**Presentation:** 4 excellent
**Contribution:** 3 good
**Rating:** 8
**Confidence:** 4

**Summary:**

This paper proposes to fine-tune smaller open-source LLMs (LIama) based on data augmentation from large closed-source LLMs (GPT-3.5). A set of data augmentation techniques are employed: answer augmentation, question bootstrapping by rephrasing, and backward reasoning, including self-verification and FOBAR. The data augmentation is applied to the GSM8K and MATH datasets. The augmented MetaMathQA dataset is then used to fine-tune the LIama model series.

Experiments on the fine-tuned 7B, 13B, and 70B LIama models demonstrate significant improvements over various baselines. The authors also made insightful analyses regarding how the perplexity and diversity of the training data affect performance, the reversal mathematical ability, reasoning paths with incorrect answers, as well as data quantity.

**Strengths:**

1. The proposed MetaMathQA dataset will be a very valuable contribution to the community.
2. The proposed data augmentation techniques achieve good performances compared to various baselines.
3. The authors made insightful analyses regarding different factors affecting the performance of such small LM fine-tuning. This analysis will not only contribute to the specific topic of mathematical reasoning but also will help the general direction of small LM fine-tuning as well.

**Weaknesses:**

1. Some baseline approaches to compare are missing, e.g., [1, 2] and code-based LLMs like [3]
2. The ablation study is not comprehensive enough. Only the 7B model is tested. Table 3 is confusing - should add a line breaker between SFT and MetaMath.

[1] MAmmoTH: Building Math Generalist Models through Hybrid Instruction Tuning, Xiang Yue, Xingwei Qu, Ge Zhang, Yao Fu, Wenhao Huang, Huan Sun, Yu Su, Wenhu Chen, 2023

[2] Platypus: Quick, Cheap, and Powerful Refinement of LLMs, Ariel N. Lee, Cole J. Hunter, Nataniel Ruiz, 2023

[3] Code Llama: Open Foundation Models for Code, Rozière et al., 2023

**Questions:**

N/A

---

> ### Author Response · Authors · 2023-11-18
> **Response to Reviewer AmB6**
>
> We would like to sincerely thank the reviewer for the useful comments on our work. We take every comment seriously and hope our response can address the reviewer’s concerns. If there are any remaining questions, we are more than happy to address them.
>
> > Q1. **Some baseline approaches to compare are missing, e.g., MAmmoTH, Platypus and Code LLaMA.**
>
> **A1.** As suggested, we added the mentioned baselines in Table 2 of the updated paper and the additional comparisons are shown in the table below. As can be seen, MetaMath outperforms other CoT-based models and Code-LLaMA with the same number of parameters.
>
> |     | \#params |  GSM8K | MATH |
> |  :---- |  :----: |  :----: | :----: |
> | Code-LLaMA | 7B | 25.2 | 13.0  |
> | MAmooTH-CoT | 7B | 50.5  | 10.4   |
> | MetaMath |7B|   66.5 | 19.8    |
> |  ||    |      |
> | Code-LLaMA |13B | 36.1 | 16.4 |
> | Platypus | 13B |  25.7 |  2.5   |
> | MAmooTH-CoT | 13B|  56.3  |  12.9  |
> | MetaMath | 13B | 72.3  |  22.4  |
> |  ||    |      |
> | Platypus | 70B | 70.6 | 15.6  |
> | MAmooTH-CoT | 70B |   72.4  | 21.1  |
> | MetaMath | 70B | 82.3 |  26.6    |
>
>
> ---
>
> > Q2. **The ablation study is not comprehensive enough. Only the 7B model is tested.**
>
> **A2.** As suggested, we conducted an ablation study on LLaMA-2-13B and the results are shown below. The observation is consistent with our ablation study on the 7B model in Section 4.3: Combing answer augmentation and rephrasing augmentation data for finetuning leads to a slightly higher accuracy. The accuracy can be further improved by merging the FOBAR and SV augmentation data. We added the results in the updated paper (Table 9 in Appendix A.7).
>
> | Method    | AnsAug | Rephrasing |  SV | FOBAR | GSM8K | MATH|
> |  :---- |  :----: | :----: | :----: | :----: |   :----: | :----: |
> | SFT       |  &#10005; |  &#10005;  |  &#10005;  |  &#10005;   |50.9 |4.5  |
> | MetaMath  |  &#10003; |  &#10005;  |   &#10005;   |  &#10005; |66.0 |5.5  |
> | MetaMath  |  &#10005; |  &#10003;  |   &#10005;   |  &#10005; |67.5 |5.9  |
> | MetaMath  |  &#10003; |  &#10003;  |   &#10005;   |  &#10005; |68.1 |5.8  |
> | MetaMath  |  &#10003; |  &#10003;  |   &#10003;   |  &#10003; |72.3 |7.2  |
>
>
> ---
>
> > Q3. **Table 3 is confusing - should add a line breaker between SFT and MetaMath**
>
> **A3.** Thanks for your suggestion and we have fixed it accordingly in the updated version.

---

### Official Review · Reviewer_pSoC · 2023-11-03

**Soundness:** 3 good
**Presentation:** 4 excellent
**Contribution:** 3 good
**Rating:** 8
**Confidence:** 4

**Summary:**

The authors propose a method for data augmentation to train LLMs for improving mathematical reasoning.
The authors combine several existing techniques such as question re-writing, self-verification, forward-backward reasoning, and answer augmentation to create a larger dataset called MetaMathQA.
The paper shows that this dataset can be distilled back into the model resulting in a fine-tuned model that outperforms several baselines on two benchmarks of mathematical reasoning.

**Strengths:**

- The proposed approach for bootstrapping seems sound and also results in better mathematical reasoning performance through thorough experimentation
- The authors also perform ablations that show that all of the bootstrapping techniques help improve performance
- The paper is well presented and easy to follow

**Weaknesses:**

- The major weakness I see is the lack of novelty. The paper in essence combines existing methods for bootsrapping.

Nevertheless, I feel that the empirical findings of the paper would be interesting to the community and therefore vote for acceptance

**Questions:**

- It is interesting that even reasoning paths with incorrect answers can be useful. Do you try to train using both correct and incorrect reasoning paths? Does this perform better than just correct?

---

> ### Author Response · Authors · 2023-11-18
> **Response to Reviewer pSoC -- Part 1**
>
> We would like to sincerely thank the reviewer for the useful comments on our work. We take every comment seriously and hope our response can address the reviewer’s concerns. If there are any remaining questions, we are more than happy to address them.
>
> > Q1. **The major weakness I see is the lack of novelty. The paper in essence combines existing methods for bootsrapping.**
>
> **A1.** This paper proposes a novel question bootstrapping idea to augment the training dataset. Compared with previous methods (e.g., SFT, RFT, WizardMath), our MetaMath introduces two novel augmentation methods:
>
> (i) **Rephrasing questions.** Existing methods (e.g., RFT and WizardMath) focus on enlarging the answer data, which can be achieved by sampling more answers from LLMs. MetaMath proposes to create more questions, which is more challenging. To the best of our knowledge, **we are the first to use a rephrasing prompting method for augmenting questions**.
>
> (ii) **Question bootstrapping by backward reasoning.**  Backward reasoning ability is crucial to solving many mathematical questions. However, the training set of mathematical tasks (e.g., GSM8K) lacks such data to improve backward reasoning. To deal with this problem, we introduce the templates (i.e., masking a number in the question and asking the LLM to predict the masked number) proposed in Self-Verification and FOBAR to create backward questions. **Note that both Self-Verification and FOBAR use backward reasoning for verification rather than data augmentation.**
>
> In summary, our paper proposes rephrasing questions and bootstrapping by backward reasoning to augment a diverse dataset --- MetaMathQA. MetaMath, which finetunes from state-of-the-art open-source LLMs on our MetaMathQA dataset, demonstrating excellent elementary mathematical problem-solving capability. In addition, we have released the MetaMathQA dataset for public use to improve the forward and backward reasoning capabilities of the LLM.

---

> > ### Author Response · Authors · 2023-11-18
> > **Response to Reviewer pSoC -- Part 2**
> >
> > > Q2. **Do you try to train using both correct and incorrect reasoning paths? Does this perform better than just correct?**
> >
> > **A2.** As suggested, we conducted an additional experiment on GSM8K to compare the performance of LLaMA-2 (7B) finetuned on (i) correct reasoning paths and (ii) correct+incorrect reasoning paths. The table below shows the testing accuracy. As can be seen, **finetuning on the correct paths is better than data containing incorrect reasoning paths.**
> >
> > |     | Accuracy|
> > |  :---- |  :----:   |
> > | Correct (20K)   |  56.2     |
> > |Correct (20K) + Incorrect (7K) | 51.4 |

---

> > > ### Comment · Reviewer_pSoC · 2023-11-21
> > > **Thanks for the additional experiments**
> > >
> > > Thanks for your response and additional experiments. I am more confident in my assessment now.
> > >
> > > Do you have any intuitions into why the incorrect paths improve accuracy but when used in tandem with the correct paths are more detrimental?

---

> > > > ### Author Response · Authors · 2023-11-23
> > > > **Response to further comments**
> > > >
> > > > Thanks for your further comments.
> > > >
> > > > > Q3. Do you have any intuitions into why the incorrect paths improve accuracy but when used in tandem with the correct paths are more detrimental?
> > > >
> > > > **A3.**
> > > > We denote three types of data used in the experiment (Section 4.8 in the submission) by
> > > > - `GSM8K-train (7K)`: the answers are correct, while data quality  is a bit low
> > > > - `MetaMathQA(GSM8K)-incorrect (7K)`: high-quality data with incorrect answers but may contain correct intermediate steps
> > > > - `MetaMathQA(GSM8K)-correct (20K)`: high-quality data with correct answers
> > > >
> > > > **The LLM trained on MetaMathQA(GSM8K)-incorrect performs better than that trained on GSM8K-train**: Though the final answer is incorrect, some intermediate steps are correct (see Example 4.1 in the submission), which may still be useful supervision signals and be more effective in training than GSM8K-train.
> > > >
> > > > However, we observe that **combining MetaMathQA-incorrect with MetaMath-correct  hurt performance**, we hypothesize this is because:
> > > > - **interpolated performance**: The performance when merging the two datasets is between those of using one of the original datasets alone. For example:
> > > >     - The performance of Correct (20K) + Incorrect (7K) is between those of Incorrect (7K) and Correct (20K).
> > > >     - The performance of Correct (20K) + GSM8K (7K) is between those of GSM8K (7K) and Correct (20K).
> > > > - **conflict labels**: when merging the correct and incorrect data, for a question $q$, it may have both correct and incorrect answers, say $a^{(+)}$ and $a^{(-)}$, respectively. During training, the gradients computed from samples $(q, a^{(+)})$ and $(q, a^{(-)})$ may conflict, hurting the training procedure and generalization ability. An observation that may reflect this hypothesis is:
> > > >     - Correct (20K) + GSM8K (7K) is better than Correct (20K) + Incorrect (7K), but GSM8K (7K) is worse than Incorrect (7K).
> > > >
> > > > |     | Accuracy|
> > > > |  :---- |  :----:   |
> > > > |GSM8K (7K) |41.6|
> > > > | Correct (20K)   |  56.2     |
> > > > | Incorrect (7K) | 43.6 |
> > > > |Correct (20K) + Incorrect (7K) | 51.4 |
> > > > |Correct (20K) + GSM8K (7K)| 54.3 |
> > > >
> > > > Thank you again for your efforts and time in improving our work.

---

### Official Review · Reviewer_nyhB · 2023-11-05

**Soundness:** 3 good
**Presentation:** 3 good
**Contribution:** 3 good
**Rating:** 8
**Confidence:** 3

**Summary:**

This paper proposes MetaMath, a fine-tuned language model specializing in mathematical reasoning. The proposed method includes bootstrapping mathematical questions by rewriting them from multiple perspectives to create the new dataset MetaMathQA. The LLaMA-2 models are then fine-tuned on the MetaMathQA dataset. Experimental results on two popular benchmarks, GSM8K and MATH, show that MetaMath significantly outperforms other open-source large language models. The authors also introduce the concept of question diversity when creating the MetaMathQA dataset, which is important in reasoning directions, and highlight that backward reasoning questions are very helpful for large language models in understanding mathematical knowledge without memorization.

**Strengths:**

1. The proposed method of bootstrapping mathematical questions by rewriting them from multiple perspectives is novel.
2. The authors construct a new dataset, MetaMathQA, by combining forward and backward mathematical questions with augmented answers. This dataset could help the community with advancing progress in mathematical reasoning.
3. The experiments are pretty extensive in that they have compared to a lot of models/approaches. (Although there are clear weaknesses in the experiments, will discuss in the weaknesses.)
4. The paper is well-organized and clearly written, making it easy to understand the motivation behind the proposal, the method, the dataset construction, and the experiments conducted.

**Weaknesses:**

1. It is unclear how the proposed bootstrapping approach generalizes to other types of multi-hop reasoning problems.
2. The ablation of the method is not rigorously done.  It is unclear if we keep increasing the number of AnsAug, we can get similar improvement.

-------

Updated after rebuttal: The new analysis table directly comparing between AnsAug and Bootstrapping is nicely done, thanks!  And thanks for adding additional models.  I have updated the scores to reflect these improvements.

**Questions:**

I think it is necessary to show that increasing AnsAug to 395K cannot further increase the performance in order to prove the point made in the paper. I understand that this experiment can be costly, so doing this in a small scale to show the trend is good enough. I would love to see a curve on the accuracy vs. # of AnsAug and a curve on the accuracy vs # of a mixed of different augmentations.

---

> ### Author Response · Authors · 2023-11-18
> **Response to Reviewer nyhB -- Part 1**
>
> We would like to sincerely thank the reviewer for the valuable comments on our work. We take every comment seriously and hope our response can address the reviewer’s concerns. If there are any remaining questions, we are more than happy to address them.
>
> > Q1: **It is unclear how the proposed bootstrapping approach generalizes to other types of multi-hop reasoning problems.**
>
> **A1.** The core idea of the proposed bootstrapping approach is to diversify the questions in both forward and backward reasoning directions. Our approach can be extended to other reasoning tasks. We conducted an additional experiment to show a successful application of our bootstrapping method to the [Game of 24](https://en.wikipedia.org/wiki/24_(puzzle)), which involves multi-hop reasoning steps to reach 24 given 4 numbers.
> Given an original question with 4 numbers (2,3,4,12), its answer (2\*3 - 4) \* 12 is an expression that can reach 24.
> We can apply answer augmentation and question bootstrapping to generate more question-answer pairs.
>
> **Answer augmentation.** The solutions of obtaining 24 given 4 numbers may not be unique, e.g., 2\*12\*(4-3) = 24 is another solution to the given numbers (2,3,4,12). Hence, for a question with 4 numbers, we enumerate all the correct solutions and obtain the answer augmentation data.
>
> **Question bootstrapping.** Game of 24 can be extended to **Game of $n$**, i.e., given 4 numbers (one number is 24), the goal is to obtain $n$ using basic arithmetic operations (+, -, \*, /). We use Game of n for question bootstrapping. We replace a number in the original question with 24 and the question is to obtain the number. This idea is similar to creating backward questions in our paper, i.e., masking a number in the question and asking the LLM to predict the number. For a Game of 24 question, we can bootstrap it and obtain 4 Game of $n$ questions, as an example shown in the table below.
>
> |          | Bootstrapping1 | Bootstrapping2 | Bootstrapping3|Bootstrapping4|
> |  :----   | :----: | :----: | :----: | :----: |
> | Input (4 numbers)     | 24, 3, 4, 12   |  2, 24, 4, 12 | 2, 3, 24, 12  | 2, 3, 4, 24  |
> | Target (n)  | 2 | 3 | 4 | 12|
> | Solution (Valid Expression)      |     (4-3)/(12/24) = 2   | (24/12+4)/2 = 3| 24/12\*3-2 = 4 | (24/4-2)\*3 = 12 |
>
> **Game of 24 Setup.** We randomly select 1362 Game of 24 questions from 4num.com,
> where 681 questions are for training and the remaining 681 questions are held-out for testing.
> We apply the above augmentation methods to generate more training data from the 681 questions. We apply answer augmentation by enumerating all the correct forward solutions and obtain an AnsAug dataset consisting of 6052 question-answer pairs. We apply question bootstrapping to obtain a bootstrapping dataset (consisting of 2724 Game of n question-answer pairs). To verify the effectiveness of the bootstrapping approach, we randomly sample 4000 question-answer pairs (Game of 24) from the AnsAug datasets, and 2052 backward question-answer pairs (Game of n) from the bootstrapping dataset. We finetune LLaMA-2-7B on AnsAug and the mixed data separately for comparison.
>
> **Results on Game of 24.** Table below shows the testing accuracy. As can be seen, our proposed augmentation approaches (AnsAug and AnsAug+Bootstrapping) have higher accuracy than SFT, which trains on the original 681 question-answer pairs. Furthermore, using question bootstrapping for augmentation can boost the performance of AnsAug. Hence, the proposed bootstrapping method is useful for Game of 24.
>
> |            |  #Samples      | Accuracy  |
> |  :----     |:----:     |  :----:   |
> |SFT | 681 | 1.8 |
> |AnsAug | 6052 | 10.2 |
> |AnsAug + Bootstrapping | 6052 | **12.0** |
>
> **Results on Game of n.** For each question-answer pair in the testing set of **Game of 24**, we create 4 more testing questions of **Game of n** using the above question bootstrapping method. In total, we obtain 3405 testing questions. Table below shows the testing accuracy. Again, using our augmentation methods performs better than SFT by a large margin. Furthermore, AnsAug + Bootstrapping performs the best, demonstrating our proposed method is also useful for Game of n.
>
> |            |  #Samples      | Accuracy  |
> |  :----     |:----:     |  :----:   |
> |SFT | 681 | 0.8 |
> |AnsAug | 6052 | 3.0 |
> |AnsAug + Bootstrapping | 6052 | **8.1** |
>
> We have included all the above experiments in Appendix A.4 of the updated paper.

---

> > ### Author Response · Authors · 2023-11-18
> > **Response to Reviewer nyhB -- Part 2**
> >
> > > Q2. **The ablation of the method is not rigorously done. It is unclear if we keep increasing the number of AnsAug, we can get similar improvement.I think it is necessary to show that increasing AnsAug to 395K cannot further increase the performance in order to prove the point made in the paper. I understand that this experiment can be costly, so doing this in a small scale to show the trend is good enough. I would love to see a curve on the accuracy vs. # of AnsAug and a curve on the accuracy vs # of a mixed of different augmentations.**
> >
> >
> > **A2.** Please note that the total number of MetaMathQA-GSM8K is 240K (as shown in Table 1). Therefore, we increase the AnsAug data to 240K (instead of 395K). We compare the performance of LLM finetuned on AnsAug data and MetaMathQA-GSM8K with question bootstrapping. The table below (and Figure 8 in Appendix A.6 of the updated paper) shows the trend on LLaMA-2-7B.
> > As can be seen, the trend is similar to Figure 2, i.e., finetuning on AnsAug rapidly reaches a state of accuracy saturation and continually increasing AnsAug data is difficult to boost performance further. In contrast, the test accuracy, when using bootstrapped questions, continues to exhibit a steady increase when AnsAug quickly saturates.
> >
> > `LLaMA-2-7B`
> >
> > | \# Samples     | 20K | 40K | 60K | 80K | 100K | 120K | 160K | 200K | 240K |
> > |  :----         |:----:|:----:|:----:|:----:|:----:|:----:|:----:|:----:|:----:|
> > | AnsAug         |54.8  |58.1  |58.8  |59.2  |59.4  |60.2  |61.1  |60.7  |60.5  |
> > | Bootstrapping   |56.2  |59.4  |61.1  |62.2  |63.0  |62.9  |65.5  |65.9  |65.8  |
> >
> > In addition, we conducted additional experiments on LLaMA-2-13B and Mistral-7B [1] for further analysis. The experimental results are shown in following tables (also Figures 9 and 10 in Appendix A.6 of the updated paper).
> > From the below tables, We can see that:
> >
> > - For models with different scales and series, the proposed bootstrapping approach consistently surpasses the baseline method (AnsAug) by a large margin, showing question bootstrapping is effective in data augmentation.
> > - When the AnsAug data are abundant, bootstrapping is more effective. For instance, when there are only 20K AnsAug data for training, bootstrapping surpasses AnsAug by 1.4%, 1.7%, 0.8% for LLaMA-2-7B, LLaMA-2-13B and Mistral-7B, respectively. For the 160K AnsAug data case, Bootstrapping surpasses AnsAug by 4.4%, 4.8%, 6.4% for LLaMA-2-7B, LLaMA-2-13B and Mistral-7B, respectively.
> >
> > `LLaMA-2-13B`
> > | \# Samples      | 20K | 40K | 80K | 120K | 160K |
> > |  :----          |:----:|:----:|:----:|:----:|:----:|
> > | AnsAug          |62.1  |64.8  |66.0  |66.6  | 67.0 |
> > | Bootstrapping   |63.8  |66.2  |68.5  |70.5  | 71.8 |
> >
> >
> > `Mistral-7B`
> > | \# Samples     | 20K | 40K | 80K | 120K | 160K |
> > |  :----         |:----:|:----:|:----:|:----:|:----:|
> > | AnsAug         |66.5  |67.1  |67.9  |68.9  |68.4  |
> > | Bootstrapping   |67.3  |69.8  |71.2  |72.1  |74.8  |
> >
> >
> > [1] Jiang, Albert Q., et al. "Mistral 7B." arXiv preprint arXiv:2310.06825 (2023).

---

> > > ### Author Response · Authors · 2023-11-21
> > > **Any additional question or concern**
> > >
> > > Dear Reviewer nyhB,
> > >
> > > We would like to thank you again for your detailed reviews. We hope that we have satisfactorily addressed your concerns.
> > >
> > > Given the limited time for discussion and that your current score is 5 we would be grateful if you could let us know whether our response has addressed your concerns or if you still have any other questions.
> > >
> > > We would be happy to do any follow-up discussion or address any additional comments.
> > >
> > > Respectfully,
> > >
> > >
> > > Authors

---

### Author Response · Authors · 2023-11-18
**General Response to Reviewers and AC**

Dear Reviewers and AC,

We sincerely thank all the reviewers and ACs for spending time on our submission. We sincerely appreciate all the efforts from the reviewers and ACs put into improving our paper. We have responded to every raised concern and hope our response can address them. We also conducted all the requested experiments and analyzed the results. Please let us know if there are any remaining questions; we are more than happy to address them.

We updated our paper with the following revisions based on the reviewers’ suggestions.

-  We added experiments of the Game of 24 and the Game of n to verify the effectiveness of bootstrapping on other multi-hop reasoning tasks (Appendix A.4).
- We added experiments on LLaMA-2-13B, Mistral-7B, and Llemma-7B to verify that MetaMathQA can generalize to other base models (Appendix A.5).
- We increased the AnsAug data to 240K and conducted experiments to show the trends of testing accuracy w.r.t. number of data in AnsAug and bootstraping data. Results show that finetuning on AnsAug rapidly reaches a state of saturation, while finetuning on bootstrapping data can increase to much higher accuracy. Moreover, we conducted additional experiments to show this trend also appears in other base models (LLaMA-2-13B, Mistral-7B) (Appendix A.6).
- We added an experiment to show that combining correct and incorrect data for finetuning is better than using incorrect data alone.
- We added the baselines (e.g., Code-LLaMA, MAmooTH-CoT, Platypus, and Orca-Platypus) suggested by reviewers to Table 2 of our paper.
- We added an experiment on GSM8K to study the effect of augmentations on a larger model LLaMA-2-13B (Appendix A.7).
- We added an experiment on the DROP dataset using zero-shot evaluation to show that MetaMath does not suffer from the issue of benchmark hacking (Appendix A.8).


Thanks again for all the effort and time.

Best,

Authors

---

### Author Response · Authors · 2023-11-21
**General Response to Reviewers and AC**

Dear Reviewers and AC,

We again express our deep gratitude to all the reviewers and AC for spending the time and efforts on our submission.
As we are nearing the end of the rebuttal period, we would greatly appreciate it if any remaining questions or concerns could be posted earlier, such that we have enough time to address them.

We sincerely hope that our responses can address the concerns raised by the reviewers.

Respectfully,

Authors

---

### Meta-Review · Area_Chair_pTqe · 2023-12-11

**Metareview:**

The authors propose a method for data augmentation to train LLMs for improving mathematical reasoning. The authors combine several existing techniques such as question re-writing, self-verification, forward-backward reasoning, and answer augmentation to create a larger dataset called MetaMathQA using GPT-3.5-turbo. The paper shows that this dataset can be used to train Llama that outperforms several baselines on two benchmarks of mathematical reasoning.

All reviewers voted 8 so a clear and strong accept.

Reviewers most liked the generated MetaMathQA dataset and how it was generated with multiple perspectives. However, the main method consists of generating a dataset tailored to the testing data, one reviewer was concerned that this will improve the specific datasets tuned on instead of more general improvements to math abilities. The authors partly addressed this in appendix A.8 with DROP results. If the authors are able to fully address this point, then they are encouraged to include it in the main results and explain how/why they succeeded where others did not, making it a more ground breaking paper. The effectiveness of similar generating in-domain data approach is also well-known in other domains with similarly impressive gains on the target but without more general improvements (instruction following, code). So the AC recommends spotlight given the strong reviews but not even higher.

**Justification For Why Not Higher Score:**

in meta review, the general approach is well-tested by now but proven not to work more generally.

**Justification For Why Not Lower Score:**

strong and consistent reviews

---

### Decision · Program_Chairs · 2024-01-16

Accept (spotlight)